# EnGAN: Latent Space MCMC and Maximum Entropy Generators for Energy-based Models

## Abstract

Unsupervised learning is about capturing dependencies between variables and driven by the contrast between the probable vs improbable configurations of these variables, often either via a generative model which only samples probable ones or with an energy function (unnormalized log-density) which is low for probable ones and high for improbable ones. Here we consider learning both an energy function and an efficient approximate sampling mechanism for the corresponding distribution. Whereas the critic (or discriminator) in generative adversarial networks (GANs) learns to separate data and generator samples, introducing an entropy maximization regularizer on the generator can turn the interpretation of the critic into an energy function, which separates the training distribution from everything else, and thus can be used for tasks like anomaly or novelty detection.

This paper is motivated by the older idea of sampling in latent space rather than data space because running a Monte-Carlo Markov Chain (MCMC) in latent space has been found to be easier and more efficient, and because a GAN-like generator can convert latent space samples to data space samples. For this purpose, we show how a Markov chain can be run in latent space whose samples can be mapped to data space, producing better samples. These samples are also used for the negative phase gradient required to estimate the log-likelihood gradient of the data space energy function. To maximize entropy at the output of the generator, we take advantage of recently introduced neural estimators of mutual information. We find that in addition to producing a useful scoring function for anomaly detection, the resulting approach produces sharp samples (like GANs) while covering the modes well, leading to high Inception and Frchet scores.

## 1 Introduction

The early work on deep learning relied on unsupervised learning (Hinton et al., 2006; Bengio et al., 2007; Larochelle et al., 2009) to train energy-based models (LeCun et al., 2006), in particular Restricted Boltzmann Machines, or RBMs. However, it turned out that training energy-based models without an analytic form for the normalization constant is very difficult, because of the challenge of estimating the gradient of the partition function, also known as the negative phase part of the log-likelihood gradient (described in more details below, Sec. 2). Several algorithms were proposed for this purpose, such as Contrastive Divergence (Hinton, 2000) and Stochastic Maximum Likelihood (Younes, 1998; Tieleman, 2008), relying on Monte-Carlo Markov Chains (MCMC) to iteratively sample from the energy-based model. However, because they appear to suffer from either high bias or high variance (due to long mixing times), training of RBMs and other Boltzmann machines has not remained competitive after the introduction of variational auto-encoders (Kingma & Welling, 2014) and generative adversarial networks or GANs (Goodfellow et al., 2014).

In this paper, we revisit the question of training energy-based models, taking advantage of recent advances in GAN-related research, and propose a novel approach to training energy functions and sampling from them, called EnGAN. The main inspiration for the proposed solution is the earlier observation (Bengio et al., 2013) made on stacks of auto-encoders that sampling in latent space (and then applying a decoder to map back to data space) led to faster mixing and more efficient sampling. The authors observed that whereas the data manifold is generally very complex and curved, the corresponding distribution in latent space tends to be much simpler and flatter. This was verified visually by interpolating in latent space and projecting back to data space through the decoder,

observing that the resulting samples look like data samples (i.e., the latent space manifold is approximately convex, with most points interpolated between examples encoded in latent space also having high probability). We propose a related approach, EnGAN, which also provides two energy functions, one in data space and one in latent space. A key ingredient of the proposed approach is the need to regularize the generator (playing the role of the decoder in auto-encoders, but with no need for an encoder) so as to increase its entropy. This is needed to make sure to produce negative examples that can kill off spurious minima of the energy function. This need was first identified by Kim & Bengio (2016), who showed that in order for an approximate sampler to match the density associated with an energy function, a compromise must be reached between sampling low energy configurations and obtaining a high-entropy distribution. However, estimating and maximizing the entropy of a complex high-dimensional distribution is not trivial, and we take advantage for this purpose of very recently proposed GAN-based approaches for maximizing mutual information (Belghazi et al., 2018; Oord et al., 2018; Hjelm et al., 2018), since the mutual information between the input and the output of the generator is equal to the entropy at the output of the generator.

In this context, the main contributions of this paper are the following:

- proposing EnGAN, a general architecture, sampling and training framework for energy functions, taking advantage of an estimator of mutual information between latent variables and generator output and approximating the negative phase samples with MCMC in latent space,

- showing that the resulting energy function can be successfully used for anomaly detection, improving on recently published results with energy-based models,

- showing that EnGAN produces sharp images - with competitive Inception and Frechet scores - and which also better cover modes than standard GANs and WGAN-GPs, while not suffering from the common blurriness issue of many maximum likelihood generative models.

## 2 LIKELIHOOD GRADIENT ESTIMATOR FOR ENERGY-BASED MODELS AND DIFFICULTIES WITH MCMC-BASED GRADIENT ESTIMATORS

Let $\boldsymbol{x}$ denote a sample in the data space $\mathcal{X}$ and $E_\theta : \mathcal{X} \to \mathbb{R}$ an energy function corresponding to minus the logarithm of an unnormalized density density function

$$p_\theta(\boldsymbol{x}) = \frac{e^{-E_\theta(\boldsymbol{x})}}{Z_\theta} \propto e^{-E_\theta(\boldsymbol{x})} \tag{1}$$

where $Z_\theta := \int e^{-E_\theta(\boldsymbol{x})} d\boldsymbol{x}$ is the partition function or normalizing constant of the density sample in the latent space. Let $p_D$ be the training distribution, from which the training set is drawn. Towards optimizing the parameters $\theta$ of the energy function, the maximum likelihood parameter gradient is

$$\frac{\partial \mathbb{E}_{\boldsymbol{x} \sim p_D}[\log p_\theta(\boldsymbol{x})]}{\partial \theta} = \mathbb{E}_{\boldsymbol{x} \sim p_D}\left[\frac{\partial E_\theta(\boldsymbol{x})}{\partial \theta}\right] - \mathbb{E}_{\boldsymbol{x} \sim p_\theta(\boldsymbol{x})}\left[\frac{\partial E_\theta(\boldsymbol{x})}{\partial \theta}\right] \tag{2}$$

where the second term is the gradient of $\log Z_\theta$, and the sum of the two expectations is zero when training has converged, with expected energy gradients in the positive phase (under the data $p_D$) matching those under the negative phase (under $p_\theta(\boldsymbol{x})$). Training thus consists in matching the shape of two distributions: the positive phase distribution (associated with the data) and the negative phase distribution (where the model is free-running and generating configurations by itself). This observation has motivated the pre-GAN idea presented by Bengio (2009) that "model samples are negative examples" and a classifier could be used to learn an energy function if it separated the data distribution from the model's own samples. Shortly after introducing GANs, Goodfellow (2014) also made a similar connection, related to noise-contrastive estimation (Gutmann & Hyvarinen, 2010). One should also recognize the similarity between Eq. 2 and the objective function for Wasserstein GANs or WGAN (Arjovsky et al., 2017). In the next section, we examine a way to train what appears to be a particular form of WGAN that makes the discriminator compute an energy function.

The main challenge in Eq. 2 is to obtain samples from the distribution $p_\theta$ associated with the energy function $E_\theta$. Although having an energy function is convenient to obtain a score allowing to compare the relative probability of different $\boldsymbol{x}$'s, it is difficult to convert an energy function into a generative function. The commonly studied approaches for this are based on Monte-Carlo Markov chains, in

which one iteratively updates a candidate configuration, until these configurations converge in distribution to the desired distribution $p_\theta$. For the RBM, the most commonly used algorithms have been Contrastive Divergence (Hinton, 2000) and Stochastic Maximum Likelihood (Younes, 1998; Tieleman, 2008), relying on the particular structure of the RBM to perform Gibbs sampling. Although these MCMC-based methods are appealing, RBMs (and their deeper form, the deep Boltzmann machine) have not been competitive in recent years compared to autoregressive models (van den Oord et al., 2016), variational auto-encoders (Kingma & Welling, 2014) and generative adversarial networks or GANs (Goodfellow et al., 2014).

What has been hypothesized as a reason for the poorer results obtained with energy-based models trained with an MCMC estimator for the negative phase gradient is that running a Markov chain in data space is fundamentally difficult when the distribution is concentrated (e.g, near manifolds) and has many modes separated by vast areas of low probability. This mixing challenge is discussed by Bengio et al. (2013) who argue that a Markov chain is very likely to produce only sequences of highly probable configurations. If two modes are far from each other and only local moves are possible (which is typically the case with MCMCs), it becomes exponentially unlikely to traverse the 'desert' of low probability which can separate two modes. This makes mixing between modes difficult in high-dimensional spaces with strong concentration of probability mass in some places (e.g. corresponding to different categories) and very low probability elsewhere. In the same papers, the authors propose a heuristic method for jumping between modes, based on performing the random walk not in data space but in the latent space of an auto-encoder. Data samples can then be obtained by mapping the latent samples to data space via the decoder. They argue that auto-encoders tend to flatten the data distribution and bring the different modes closer to each other. The EnGAN sampling method proposed here is highly similar but leads to learning both an energy function in data space and one in latent space, from which we find that better samples are obtain. The energy function can be used to perform the appropriate Metropolis-Hastings rejection. Having an efficient way to approximately sample from the energy function also opens to the door to estimating the log-likelihood gradient with respect to the energy function according to Eq. 2, as outlined below.

## 3 Turning GAN Discriminators into Energy Functions with Entropy Maximization

Turning a GAN discriminator into an energy function has been studied in the past (Kim & Bengio, 2016; Zhao et al., 2016; Dai et al., 2017) but in order to turn a GAN discriminator into an energy function, a crucial and difficult requirement is the maximization of entropy at the output of the generator. Let's see why. In Eq. 2, we can replace the difficult to sample $p_\theta$ by another generative process, say $p_G$, such as the generative distribution associated with a GAN generator:

$$\mathcal{L}_E = \mathbb{E}_{\boldsymbol{x} \sim p_D} \left[ \frac{\partial E_\theta(\boldsymbol{x})}{\partial \theta} \right] - \mathbb{E}_{\boldsymbol{x} \sim p_G(\boldsymbol{x})} \left[ \frac{\partial E_\theta(\boldsymbol{x})}{\partial \theta} \right] + \Omega \tag{3}$$

where $\Omega$ is a regularizer which we found necessary to avoid numerical problems in the scale (temperature) of the energy. In this paper we use a gradient norm regularizer (Gulrajani et al., 2017) $\Omega = \left|\left| \nabla_{\boldsymbol{x}} E_\theta(\boldsymbol{x}) \right|\right|^2$ for this purpose. This is similar to the training objective of a WGAN as to Eq. 2, but this interpretation allows to train the energy function only to the extent that $p_G$ is sufficiently similar to $p_\theta$. To make them match, consider optimizing $G$ to minimize the KL divergence $KL(p_G||p_\theta)$, which can be rewritten in terms of minimizing the energy of the samples from the generator while maximizing the entropy at the output of the generator: $KL(p_G||p_\theta) = H[p_G] - E_{p_G}[\log p_\theta(\boldsymbol{x})]$ as already shown by Kim & Bengio (2016). When taking the gradient of $KL(p_G||p_\theta)$ with respect to the parameters $\boldsymbol{w}$ of the generator, the partition function of $p_G$ disappears and we equivalently can optimize $\boldsymbol{w}$ to minimize

$$\mathcal{L}_G = -H[p_G] + \mathbb{E}_{\boldsymbol{z} \sim p_z} E_\theta(G(\boldsymbol{z})) \tag{4}$$

where $p_z$ is the prior distribution of the latent variable of the generator.

In order to maximize the entropy at the output of the generator, we propose to exploit another GAN-derived framework in order to estimate and maximize mutual information between the input and output of the generator network. The entropy at the output of a deterministic function (the generator in our case) can be computed using an estimator of mutual information between the input and output of that function, since the conditional entropy term is 0 because the function is deterministic. With

$x = G(z)$ the function of interest:

$$I(X, Z) = H(X) - H(X|Z) = H(G(Z)) - \underbrace{H(G(Z)|Z)}_{0}$$

Hence, any neural mutual information maximization method such as MINE (Belghazi et al., 2018), noise constrastive estimation (Oord et al., 2018) and DeepINFOMAX (Hjelm et al., 2018) can be applied to estimate and maximize the entropy of the generator. All these estimators are based on training a discriminator which separates the joint distribution $p(X, Z)$ from the product of the corresponding marginals $p(X)p(Z)$. As proposed by Brakel & Bengio (2017) in the context of using a discriminator to minimize statistical dependencies between the outputs of an encoder, the samples from the marginals can be obtained by creating negative examples pairing an $X$ and a $Z$ from different samples of the joint, e.g., by independently shuffling each column of a matrix holding a minibatch with one row per example. The training objective for the discriminator can be chosen in different ways. In this paper, we used the Deep INFOMAX (DIM) estimator (Hjelm et al., 2018), which is based on maximizing the Jensen-Shannon divergence between the joint and the marginal (see Nowozin et al. for the original F-GAN formulation).

$$\mathcal{I}^{JSD}(X, Z) = \mathbb{E}_{p(X,Z)}[-s_+(-T(X, Z))] - \mathbb{E}_{p(X)p(Z)}[s_+(T(X, Z))] \tag{5}$$

where $s+(a) = \log(1+e^a)$ is the softplus function. The discriminator $T$ used to increase entropy at the output of the generator is trained by maximizing $\mathcal{I}^{JSD}(X, Z)$ with respect to the parameters of $T$. With $X = G(Z)$ the output of the generator, $\mathcal{I}^{JSD}(G(Z), Z)$ is one of the terms to be minimized the objective function for training $G$, with the effect of maximizing the generator's output entropy $H(G(Z))$. The overall training objective for $G$ is

$$\mathcal{L}_G = -\mathcal{I}^{JSD}(G(Z), Z) + \mathbb{E}_{\boldsymbol{z} \sim p_z} E_\theta(G(\boldsymbol{z})) \tag{6}$$

where $Z \sim p_z$, the latent prior (typically a $N(0, I)$ Gaussian).

## 4 PROPOSED LATENT SPACE MCMC AND MAXIMUM ENTROPY GENERATOR FOR ENERGY-BASED MODELS

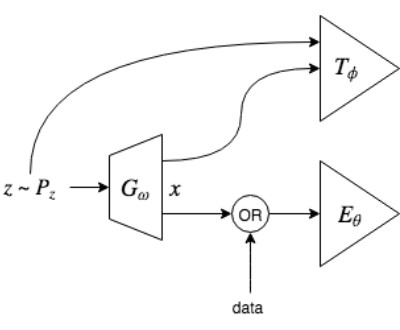

Figure 1: EnGAN model overview where $G_\omega$ is the Generator network, $T_\phi$ is the Statistics network used for MI estimation and $E_\theta$ is the energy network

One option to generate samples is simply to use the usual GAN approach of sampling a $z \sim p_z$ from the latent prior and then output $x = G(z)$, i.e., obtain a sample $\boldsymbol{x} \sim p_G$. Since we have an energy function, another option is to run an MCMC in data space, and we have tried this with both Metropolis-Hastings (with a Gaussian proposal) and adjusted Langevin (detailed below, which does a gradient step down the energy and adds noise, then rejects high-energy samples). However, we have interestingly obtained the best samples by considering $E_\theta \circ G$ as an energy function in latent space and running an adjusted Langevin in that space (compare Fig. 4 with Fig. 7.1). Then, in order to produce a data space sample, we apply $G$. For performing the MCMC sampling, we use the Metropolis-adjusted Langevin algorithm (MALA), with Langevin dynamics producing a proposal distribution in the latent space as follows:

$$\tilde{\boldsymbol{z}}_{t+1} = \boldsymbol{z}_t - \alpha \frac{\partial E_\theta(G_\omega(\boldsymbol{z}))}{\partial \boldsymbol{z}} + \epsilon \sqrt{2 * \alpha}, \text{ where } \epsilon \sim \mathcal{N}(0, I_d)$$

Next, the proposed $\tilde{z}_{t+1}$ is accepted or rejected using the Metropolis Hastings algorithm, by computing the acceptance ratio

$$r = \frac{p(\tilde{x}_{t+1})}{p(x_t)} = \exp\left\{-E_\theta(G_\omega(\tilde{z}_{t+1})) + E_\theta(G_\omega(z_t))\right\}$$

and accepting (setting $z_{t+1} = \tilde{z}_{t+1}$) with probability $r$.

The overall training procedure for EnGAN is detailed in Algorithm 1, with MALA referring to the above procedure for sampling by MCMC, with $n_{\text{mcmc}}$ steps. When $n_{mcmc}=0$, we recover the base

case where $z$ is only sampled from the prior and passed through $G$, and no MCMC is done to clean up the sample.

---

**Algorithm 1** **EnGAN Training Procedure** Default values: Adam parameters $\alpha = 0.0001, \beta_1 = 0.5, \beta_2 = 0.9; \lambda = 0.1; n_\varphi = 5$

---

**Require:** Score penalty coefficient $\lambda$, number of energy function updates $n_\varphi$ per generator updates, number of MCMC steps $n_{\text{mcmc}}$, number of training iterations $T$, Adam hyperparameters $\alpha$, $\beta_1$ and $\beta_2$.
**Require:** Energy function $E_\theta$ with parameters $\theta$, entropy statistics network $T_\phi$ with parameters $\phi$, generator network $G_\omega$ with parameters $\omega$, minibatch size $m$
  **for** $t = 1, ..., T$ **do**
    **for** $1, ..., n_\varphi$ **do**
      Sample minibatch of real data $\{\mathbf{x}^{(1)}, ..., \mathbf{x}^{(m)}\} \sim P_D$.
      Sample minibatch of latent variables $\{\mathbf{z}_0^{(1)}, ..., \mathbf{z}_0^{(m)}\} \sim P_z$.
      **for** $\tau = 1, ..., n_{\text{mcmc}}$ **do**
        $\mathbf{z}_{\tau+1} \leftarrow \text{MALA}(\mathbf{z}_\tau, E_\theta \circ G)$
      **end for**
      $\tilde{\mathbf{x}} \leftarrow G_\omega(\mathbf{z}_{n_{\text{mcmc}}})$
$$\mathcal{L}_E \leftarrow \frac{1}{m}\left[\sum_i^m E_\theta(\mathbf{x}^{(i)}) - \sum_i^m E_\theta(\tilde{\mathbf{x}}^{(i)}) + \lambda \sum_i^m \left|\left|\nabla_{\mathbf{x}^{(i)}} E_\theta(\mathbf{x}^{(i)})\right|\right|^2\right]$$
      $\theta \leftarrow \text{Adam}(\mathcal{L}_E, \theta, \alpha, \beta_1, \beta_2)$
    **end for**
    Sample minibatch of latent variables $\mathbf{z} = \{\mathbf{z}^{(1)}, ..., \mathbf{z}^{(m)}\} \sim P_z$.
    $\tilde{\mathbf{x}} \leftarrow G_\omega(\mathbf{z})$
    Per-dimension shuffle of the minibatch $\mathbf{z}$ of latent variables, obtaining $\{\tilde{\mathbf{z}}^{(1)}, ..., \tilde{\mathbf{z}}^{(m)}\}$.
$$\mathcal{L}_H \leftarrow \frac{1}{m}\sum_i^m \left[\log \sigma(T_\phi(\mathbf{x}^{(i)}, \mathbf{z}^{(i)})) - \log\left(1 - \sigma(T_\phi(\mathbf{x}^{(i)}, \tilde{\mathbf{z}}^{(i)}))\right)\right]$$
$$\mathcal{L}_G \leftarrow \frac{1}{m}\left[\sum_i^m E_\theta(\tilde{\mathbf{x}}^{(i)})\right] + \mathcal{L}_H$$
    $\omega \leftarrow \text{Adam}(\mathcal{L}_G, \omega, \alpha, \beta_1, \beta_2)$
    $\phi \leftarrow \text{Adam}(\mathcal{L}_H, \phi, \alpha, \beta_1, \beta_2)$
  **end for**
The gradient-based updates can be performed with any gradient-based learning rule. We used Adam in our experiments.

---

## 5 EXPERIMENTAL SETUP

### 5.1 SYNTHETIC TOY DATASETS

Generative models trained with maximum likelihood often suffer from the problem of spurious modes and excessive entropy of the trained distribution, where the model incorrectly assigns high probability mass to regions not present in the data manifold. Typical energy-based models such as RBMs suffer from this problem partly because of the poor approximation of the negative phase gradient, as discussed above.

To check if EnGAN suffers from spurious modes, we train the energy-based model on synthetic 2D datasets (swissroll, 25gaussians and 8gaussians) similar to Gulrajani et al. (2017) and visualize the energy function.

From the probaility density plots on Figure 1, we can see that the energy model doesn't suffer from spurious modes and learns a sharp energy distribution.

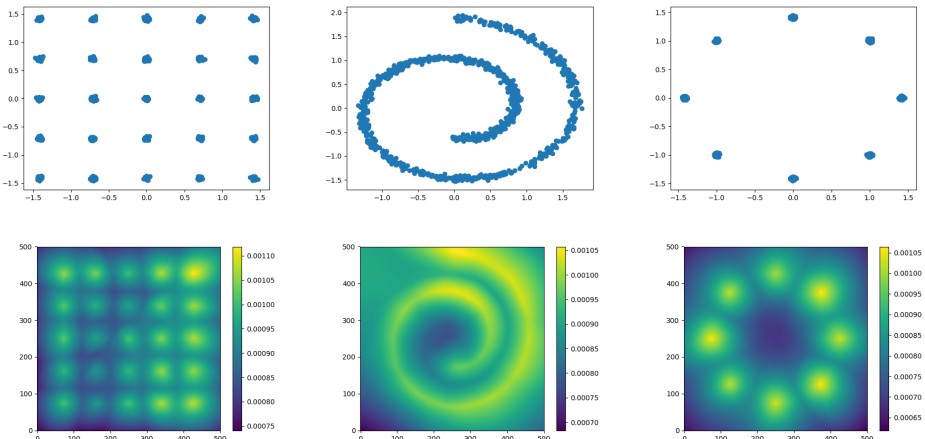

Figure 2: **Top:** Training samples for the 3 toy datasets - 25gaussians, swissroll and 8gaussians. **Bottom:** Corresponding probabiltiy density visualizations. Density was estimated using a sample based approximation of the partition function.

## 5.2 DISCRETE MODE COLLAPSE EXPERIMENT

GANs have been notoriously known to have issues with mode collapse, by which certain modes of the data distribution are not at all represented by the generative model. Similar to the mode dropping issue that occurs in GANs, our generator is prone to mode dropping as well, since it is matched with the energy model's distribution using a reverse KL penalty $D_{\mathrm{KL}}[P_G \parallel P_E]$. Although the entropy maximization term attempts to fix this issue by maximizing the entropy of the generator's distribution, it is important to verify this effect experimentally. For this purpose, we follow the same experimental setup as Metz et al. (2016) and Srivastava et al. (2017). We train our generative model on the StackedMNIST dataset, which is a synthetic dataset created by stacking MNIST on different channels. The number of modes can be counted using a pretrained MNIST classifier, and the KL divergence can be calculated empirically between the mode count distribution produced by the generative model and true data (assumed to be uniform).

Table 1: Number of captured modes and Kullback-Leibler divergence between the training and samples distributions for ALI (Dumoulin et al., 2016), Unrolled GAN (Metz et al., 2016), Vee-GAN (Srivastava et al., 2017), PacGAN (Lin et al., 2017), WGAN-GP (Gulrajani et al., 2017). Numbers except our model and WGAN-GP are borrowed from Belghazi et al. (2018)

| (Max $10^3$) | Modes | KL | (Max $10^4$) | Modes | KL |
|---|---|---|---|---|---|
| Unrolled GAN | 48.7 | 4.32 | WGAN-GP | 9538.0 | 0.9144 |
| VEEGAN | 150.0 | 2.95 | Our EnGAN | 10000.0 | 0.0480 |
| WGAN-GP | 959.0 | 0.7276 | | | |
| PacGAN | $1000.0 \pm 0.0$ | $0.06 \pm 1.0e^{-2}$ | | | |
| Our EnGAN | 1000.0 | 0.0313 | | | |

From Table 1, we can see that our model naturally covers all the modes in that data, without dropping a single mode. Apart from just representing all the modes of the data distribution, our model also better matches the data distribution as evidenced by the very low KL divergence scores as compared to the baseline WGAN-GP.

We noticed empirically that modeling $10^3$ modes was quite trivial for benchmark methods such as WGAN-GP (Gulrajani et al., 2017). Hence, we also try evaluating our model on a new dataset with $10^4$ modes (4 stacks). The 4-StackedMNIST was created to have similar statistics to the original 3-StackedMNIST dataset. We randomly sample and fix $128 \times 10^4$ images to train the generative model and take $26 \times 10^4$ samples for evaluations.

## 5.3 PERCEPTUAL QUALITY OF SAMPLES

Generative models trained with maximum likelihood have often been found to produce more blurry samples. Our energy model is trained with maximum likelihood to match the data distribution and the generator is trained to match the energy model's distribution with a reverse KL penalty. To evaluate if our generator exhibits blurriness issues, we train our EnGAN model on the standard benchmark 32x32 CIFAR10 dataset for image modeling. We additionally train our models on the 64x64 cropped CelebA - celebrity faces dataset to report qualitative samples from our model. Similar to recent GAN works (Miyato et al., 2018), we report both Inception Score (IS) and Frchet Inception Distance (FID) scores on the CIFAR10 dataset and compare it with a competitive WGAN-GP baseline.

Table 2: Inception scores and FIDs with unsupervised image generation on CIFAR-10. 50000 samples were used to compute Inception Score and FID.

| Method | Inception score | FID |
|---|---|---|
| Real data | 11.24±.12 | 7.8 |
| WGAN-GP | 6.52 ± .08 | 35.85 |
| Our model | 6.31 ± .06 | 39.01 |

From Table 2, we can see that in addition to learning an energy function, EnGAN trains generative model producing samples comparable to recent adversarial methods such as WGAN-GP (Gulrajani et al., 2017) widely known for producing samples of high perceptual quality. Additionally, we attach samples from the generator trained on the CelebA dataset and the 3-StackedMNIST dataset for qualitative inspection. As shown below in Fig. 4, the visual quality of the samples can be further improved by using the proposed MCMC sampler.

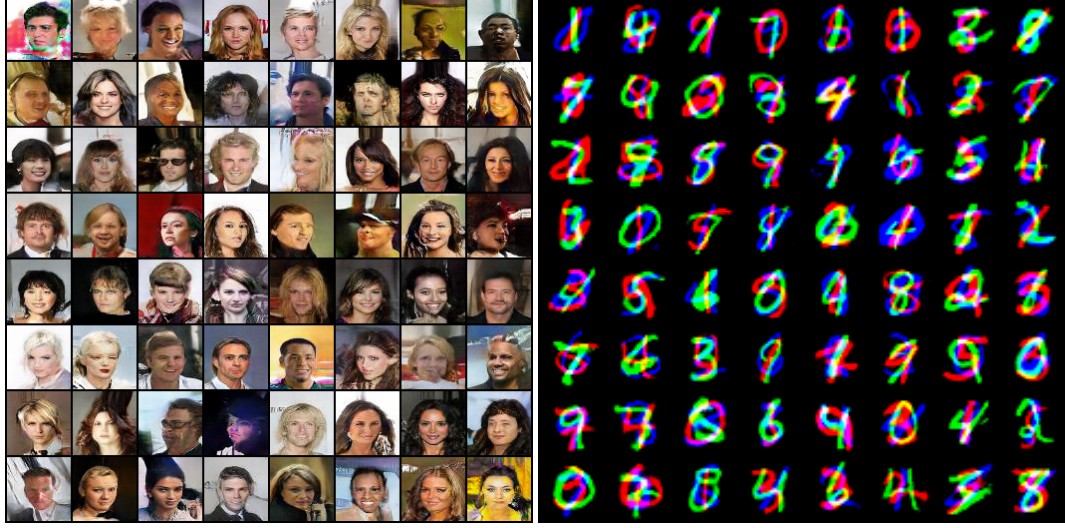

Figure 3: **Left**: 64x64 samples from the CelebA dataset **Right**: 28x28 samples from the 3-StackedMNIST dataset. All samples are produced by the generator in a single step, without MCMC fine-tuning (see Fig. 4 for that).

## 5.4 APPLICATION TO ANOMALY DETECTION

Apart from the usefulness of energy estimates for relative density estimation (up to the normalization constant), energy functions can also be useful to perform unsupervised anomaly detection. Unsupervised anomaly detection is a fundamental problem in machine learning, with critical applications in many areas, such as cybersecurity, complex system management, medical care, etc. Density estimation is at the core of anomaly detection since anomalies are data points residing in low probability

density areas. We test the efficacy of our energy-based density model for anomaly detection using two popular benchmark datasets: KDDCUP and MNIST.

**KDDCUP** We first test our generative model on the KDDCUP99 10 percent dataset from the UCI repository (Lichman et al., 2013).

Our baseline for this task is Deep Structured Energy-based Model for Anomaly Detection (DSEBM) (Zhai et al., 2016), which trains deep energy models such as Convolutional and Recurrent EBMs using denoising score matching instead of maximum likelihood, for performing anomaly detection. We also report scores on the state of the art DAGMM (Zong et al., 2018), which learns a Gaussian Mixture density model (GMM) over a low dimensional latent space produced by a deep autoencoder.

We train our model on the KDD99 data and use the score norm $||\nabla_x E_\theta(x)||_2^2$ as the decision function, similar to Zhai et al. (2016).

Table 3: Performance on the KDD99 dataset. Values for OC-SVM, DSEBM, Efficient GAN values were obtained from Zong et al. (2018). Values for our model are derived from 5 runs. For each individual run, the metrics are averaged over the last 10 epochs.

| **Model** | Precision | Recall | F1 |
|---|---|---|---|
| Kernel PCA | 0.8627 | 0.6319 | 0.7352 |
| OC-SVM | 0.7457 | 0.8523 | 0.7954 |
| DSEBM-e | 0.8619 | 0.6446 | 0.7399 |
| DAGMM | 0.9297 | 0.9442 | 0.9369 |
| Our EnGAN | **0.9307 ± 0.0146** | **0.9472 ± 0.0153** | **0.9389 ± 0.0148** |

From Table 3, we can see that our EnGAN energy function outperforms the previous SOTA energy-based model (DSEBM) by a large margin (+0.1990 F1 score) and is comparable to the current SOTA model (DAGMM) specifically designed for anomaly detection.

**MNIST** Next we evaluate our generative model on anomaly detection of high dimensional image data. We follow the same experiment setup as (Zenati et al., 2018) and make each digit class an anomaly and treat the remaining 9 digits as normal examples. We also use the area under the precision-recall curve (AUPRC) as the metric to compare models.

Table 4: Performance on the unsupervised anomaly detection task on MNIST measured by area under precision recall curve. Numbers except ours are obtained from (Zenati et al., 2018). Results for our model are averaged over last 10 epochs to account for the variance in scores.

| **Heldout Digit** | VAE | Our EnGAN | BiGAN-$\sigma$ |
|---|---|---|---|
| 1 | 0.063 | 0.281 ± 0.035 | 0.287 ± 0.023 |
| 4 | 0.337 | 0.401 ± 0.061 | 0.443 ± 0.029 |
| 5 | 0.325 | 0.402 ± 0.062 | 0.514 ± 0.029 |
| 7 | 0.148 | 0.29 ± 0.040 | 0.347 ± 0.017 |
| 9 | 0.104 | 0.342 ± 0.034 | 0.307 ± 0.028 |

From Table 4, it can be seen that our energy model outperforms VAEs for outlier detection and is comparable to the SOTA BiGAN-based anomaly detection methods for this dataset (Zenati et al., 2018) which train bidirectional GANs to learn both an encoder and decoder (generator) simultaneously. An advantage with our method is that it has theoretical justification for the usage of energy function as a decision function, whereas the BiGAN-$\sigma$ model lacks justification for using a combination of the reconstruction error in output space as well as the discriminator's cross entropy loss for the decision function.

## 5.5 MCMC SAMPLING

To show that the Metropolis Adjusted Langevin Algorithm (MALA) performed in latent space produced good samples in observed space, we attach samples from the beginning (with $z$ sampled from a Gaussian) and end of the chain for visual inspection. From the attached samples, it can be seen that the MCMC sampler appears to perform a smooth walk on the image manifold, with the initial and final images only differing in a few latent attributes such as hairstyle, background color, face orientation, etc.

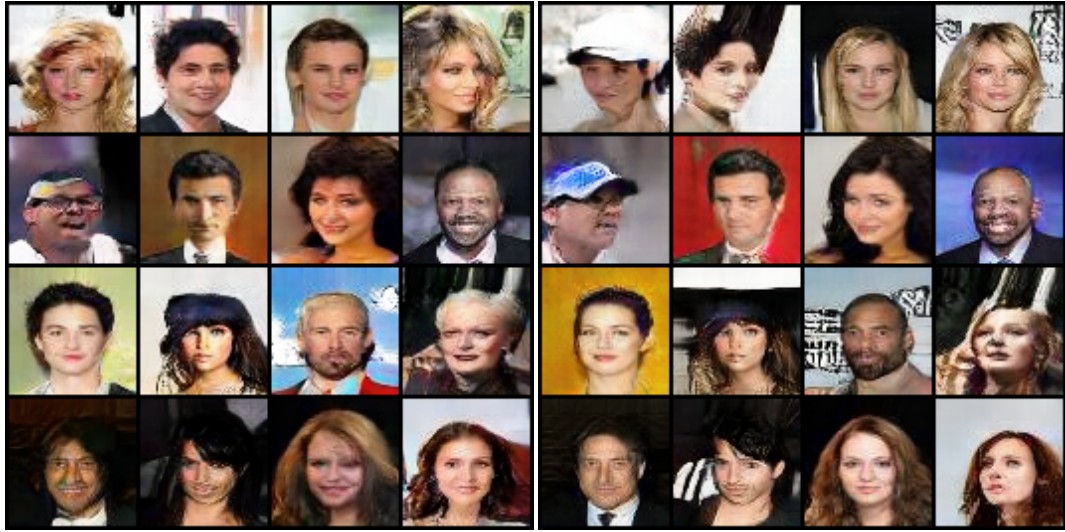

Figure 4: **Left:** Samples at the beginning of the chain (i.e. simply from the ordinary generator, $z \sim N(0, I)$). **Right:** Generated samples after 100 iterations of MCMC using the MALA sampler. We see how the chain is smoothly walking on the image manifold and changing semantically meaningful and coherent aspects of the images.

## 6 CONCLUSIONS

We proposed EnGAN, an energy-based generative model that produces energy estimates using an energy model and a generator that produces fast approximate samples. This takes advantage of novel methods to maximize the entropy at the output of the generator using a GAN-like technique. We have shown that our energy model learns good energy estimates using visualizations in toy 2D data and through performance in unsupervised anomaly detection. We have also shown that our generator produces samples of high perceptual quality by measuring Inception and Frchet scores and shown that EnGAN is robust to the respective weaknesses of GAN models (mode dropping) and maximum-likelihood energy-based models (spurious modes). We found that running an MCMC in latent space rather than in data space (by composing the generator and the data-space energy to obtain a latent-space energy) works substantially better than running the MCMC in data-space.

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

# 7 APPENDIX

## 7.1 MCMC IN DATA SPACE

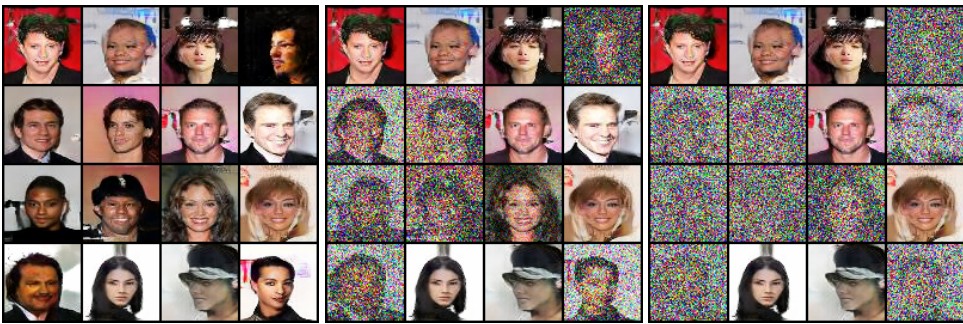

Figure 5: Samples from the beginning, middle and end of the chain performing MCMC sampling in visible space. Initial sample is from the generator ($p_G$) but degrades as we follow MALA directly in data space. Compare with samples obtained by running the chain in latent space and doing the MH rejection according to the data space energy (Fig. 4). It can be seen that MCMC in data space has poor mixing and gets attracted to spurious modes.

## 7.2 ARCHITECTURE AND HYPERPARAMETERS

For all experiments we use Adam as the optimizer with $\alpha = 0.0001, \beta_1 = 0.5, \beta_2 = 0.9$. We used $n_{\mathrm{mcmc}} = 0$ (no MCMC steps during training) for all scores reported in the paper.

**Toy Data:** The generator, energy-model and the statistics network are simple 3-hidden layer MLPs with dimensionality 512. The input to the statistics network is a conatenation of the inputs $\mathbf{x}$ and latents $\mathbf{z}$. For these experiments, we use the energy norm co-efficient $\lambda = 0.1$

**StackedMNIST:** : In line with previous work, we adopt the same architectural choices for the generator and energy-model / discriminator as VeeGAN (Srivastava et al., 2017). The statistics network is modeled similar to the energy-model, except with the final MLP which now takes as input both the latents $\mathbf{z}$ and reduced feature representation of $\mathbf{x}$ produced by the CNN.

**CIFAR10:** For the CIFAR10 experiments, we adopt the same 'Standard CNN' architecture as in SpectralNorm (Miyato et al., 2018). We adapt the architecture for the Statistics Network similar to the StackedMNIST experiments as mentioned above. For these experiments, we use the energy norm co-efficient $\lambda = 10$

**Anomaly Detection:** For the KDD99 dataset, we adopt the same architecture as (Zenati et al., 2018). We noticed that using $n_\psi = 1$ and $\lambda = 10^5$ worked best for these experiments. A large energy norm coefficient was specifically necessary since the energy model overfit to some artifacts in the data and exploded in value.

For the MNIST anomaly detection experiments, we use the same architecture as the StackedMNIST experiments.

