# OpenReview forum: "EnGAN: Latent Space MCMC and Maximum Entropy Generators for Energy-based Models"
_ICLR.cc/2019/Conference_

### Official Review · AnonReviewer2 · 2018-10-26
**Interesting approach, but not fully justified**

**Rating:** 5
**Confidence:** 4

**Review:**

Thank you for an interesting read.

The paper proposes an approximate training technique for energy-based models (EBMs). More specifically, the samples used negative phase gradient in EBM training is approximated by samples from another generator. This "approximate generator" is a composition of a decoder (which, with a Gaussian prior on latent variable z, is trained to approximate the data distribution) and another EBM in latent space. The authors show connections to WGAN training, thus the name EnGAN. Experiments on natural image generation and anomaly detection show promising improvements, although not very significant.

From my understanding of the paper, the main contribution of the paper comes from section 4, which proposes a latent-space MCMC scheme to improve sample quality. I have seen several papers fusing EBMs and GAN training together and to the best of my knowledge section 4 is novel (but with problems, see below). Section 3's recipe is quite standard, e.g. as seen in Kim and Bengio (2017), and in principle contrastive divergence also uses the same idea. The idea of estimating of the entropy term for the implicit distribution p_G with adversarial mutual information estimation is something new, although quite straight-forward.

Although I do agree that MCMC mixing in x space can be much harder than MCMC mixing in z space, since I don't think the proposed latent-space MCMC scheme is exact (apart from finite-time simulation, rejection...), I don't see theoretically why the method works.

1. The MCMC method essentially samples z from another EBM, where that EBM(z) has energy function -E_{\theta}(G(z)), and then generate x = G(z). Note here EBM(z) != p(z). The key issue is, even when p_G(x) = p_{\theta}(x), there is no guarantee that the proposed latent-space MCMC method would return x samples according to distribution p_{\theta}(x). You can easily work out a counter example by considering G is an invertible transformation. Therefore I don't understand why doing MCMC on this latent-space EBM can help improve sample quality in x space.

2. Continuing point 1, with Algorithm 1 that only fits p_G(x) towards p_{\theta}(x), I am confident that the negative phase gradient is still quite biased. Why not just use the latent-space MCMC sampler composited with G as the generator, and use these MCMC samples to train both the decoder G and the mutual information estimator?

3. I am not exactly sure why the gradient norm regulariser in (3) make sense here? True that it would be helpful to correct the bias of the negative phase, but why this particular form? We are not doing WGAN here and in general we don't usually put a Lipschitz constraint on the energy function. I've seem several GAN papers arguing that gradient penalty helps in cases beyond WGAN, but most of them are just empirical observations...
Also the Omega regulariser is computed on which x? On data? Do you know whether the energy is guaranteed to be minimized at data locations? In this is that appropriate to call Omega a regulariser?

The presentation is overall clear, although I think there are a few typos and confusing equations:

1. There should be a negative sign on the LHS of equation 2.
2. Equation 3 is inconsistent with the energy update equation in Algorithm 1. The latter one makes more sense.
3. Where is the ratio between the transition kernels in the acceptance ratio equation? In general for Langevin dynamics the transition kernel is not symmetric.

---

> ### Author Response · Authors · 2018-11-14
> **Thanks for the constructive feedback, composing the generator with the energy function gets rid of spurious modes (1/2)**
>
> We thank the reviewer for the positive and constructive feedback. We appreciate that the reviewer finds that our method is clearly explained.
>
> * “1. The MCMC method essentially samples z from another EBM, where that EBM(z) has energy function -E_{\theta}(G(z)), and then generate x = G(z). Note here EBM(z) != p(z). The key issue is, even when p_G(x) = p_{\theta}(x), there is no guarantee that the proposed latent-space MCMC method would return x samples according to distribution p_{\theta}(x). You can easily work out a counter example by considering G is an invertible transformation. Therefore I don't understand why doing MCMC on this latent-space EBM can help improve sample quality in x space.”
>
> Our hypothesis is the following: composing the generator with the energy function gets rid of spurious modes of the energy which the generator cannot represent. If the generator did sample from these spurious modes, then the negative samples from the generator would have made the energy function learn to get rid of these spurious modes, via the 2nd term of eqn 3 when training the energy function. Hence we get a cleaned-up version of the energy function. Spurious modes of the energy function which have not been eliminated via training through eqn 3 are thus erased by this composition of G with E. Now there may be a price to pay for this, i.e., G may also be missing some modes (as usual with GANs). However, because we have the entropy maximization term (eqn 4), we at least train in a way that attempts to minimize this problem. We agree that the composed energy function is different from E. The other good thing about MCMC in the composed energy function is that it seems to also be easier, following the observations of Bengio et al 2013, because the data manifold has been somewhat flattened in the latent space of the generator.
>
> * “2. Continuing point 1, with Algorithm 1 that only fits p_G(x) towards p_{\theta}(x), I am confident that the negative phase gradient is still quite biased. Why not just use the latent-space MCMC sampler composited with G as the generator, and use these MCMC samples to train both the decoder G and the mutual information estimator?”
>
> This is a good idea, which we did not execute yet because it would slow down training 10-fold, but it is an interesting direction to follow-up with.
>
> * “3. I am not exactly sure why the gradient norm regularizer in (3) make sense here? True that it would be helpful to correct the bias of the negative phase, but why this particular form? We are not doing WGAN here and in general we don't usually put a Lipschitz constraint on the energy function. I've seem several GAN papers arguing that gradient penalty helps in cases beyond WGAN, but most of them are just empirical observations...
> Also the Omega regularizer is computed on which x? On data? Do you know whether the energy is guaranteed to be minimized at data locations? In this is that appropriate to call Omega a regularizer?”
>
> The regularizer ||dEnergy(x)/dx||^2 is not just a smoothness regularizer but it also makes data points x energy minima (because ||dEnergy(x)/dx|| should be 0 at data points). This thus helps to learn a better energy function. Note that this is similar in spirit to score matching, which also carves the energy function so that it has local minima at the training points. The regularizer also stabilizes the temperature (scale) of the energy function, making training stable (avoiding continued growth of precision, inverse temperature, as training continues).
>
> * “2. Equation 3 is inconsistent with the energy update equation in Algorithm 1. The latter one makes more sense.”
>
> Sorry for the typo in eqn 3. The LHS should have been the gradient of L_E wrt theta, and Omega on the RHS should have been dOmega/dtheta.
>
> * “3. Where is the ratio between the transition kernels in the acceptance ratio equation? In general for Langevin dynamics the transition kernel is not symmetric.”
>
> The correction term to be added to -E(G(z'))+E(G(z))  (where z' = new z, and z = old z) would be:
>
> log (q(z|z')/q(z|z')) = 0.5 ( ||eps||^2 - ||eps - sqrt(alpha/2)(E'(z) - E'(z'))||^2 )
>
> where q is the proposal distribution producing z' from z and E' = gradient of E. We tried using the full formula but that it did not seem to make a discernible difference.
>
> Please let us know if anything is unclear here or if there is any other comparison that would be helpful in clarifying things more.

---

> ### Author Response · Authors · 2018-11-14
> **Addressing concern about latent-space MCMC not sampling from p_theta (2/2)**
>
> ** In general, we agree that the proposed MCMC sampling procedure is not sampling from p_theta and the energy function (see (1) why this is actually a good thing). But consider the special case (not enforced here) where G is invertible, i.e., there exists only one z such that G(z)=x for all x in the regions of interest. Let p_{EG}(x) be the density in x-space corresponding to x=G(z) and z sampled following the composed energy function E(G(z)). Then
>
> p_{EG}(x) = \propto \int 1_{G(z)=x} exp(-E(G(z)) dz
>                 = exp(-E(x)) \int 1_{G(z)=x} dz
>                 = exp(-E(x))
>
> where the first line (with 1_{} indicating a dirac delta function) comes from considering all the z's which could give rise to the given x and weighing their probability by exp(-E(G(z)) (and ignoring the partition function as we only care about the relative probabilities here), the second line comes from the observation that G(z)=x for all the non-zero integrands so we can take the exponential out of the integral at G(z)=x, and the 3rd line from integrating a dirac when there is only one point z which satisfies the condition, i.e., G is invertible.
>
> Ongoing work is investigating how to make G approximately invertible using a reconstruction loss in z-space, although an alternative would be to structure G so that it is invertible by construction, as in NICE / real NVP.

---

> > ### Comment · AnonReviewer2 · 2018-11-14
> > **The explanation is problematic :(**
> >
> > Thank you for following up my biggest concern.
> >
> > Unfortunately I don't think you can easily go away from this issue just by doing integrals. Otherwise all previous developments on normalising flows will be problematic! More specifically, they (e.g. NICE/real NVP/IAF/MAF) considered the following model:
> > p_z(z) = N(0, I), x = G(z), G is invertible
> > And you can see all of them clearly wrote p_x(x) = p_z(G^{-1}(x)) |dz/dx|.
> >
> > I would suggest reading the following relevant paper which might be helpful to clear your confusions.
> >
> > https://arxiv.org/abs/1708.01529
> >
> > Best,

---

> > > ### Author Response · Authors · 2018-11-15
> > > **Further clarification**
> > >
> > > We agree that the first line of the equations above is incorrect. Thanks for pointing it out. It works in the discrete case but in the continuous case it needs an extra factor for the determinant of G'. Note that this discussion is about future work to extend the submitted paper, so indeed, in order to encourage the density matching property we would need a regularizer (or a hard constraint) to make G not just invertible but also with G' having singular values as close to 1 as possible to preserve volume. We could use a NICE/NVP-like parametrization but the surprising thing is that experimentally the proposed method works without that hard constraining in the various settings we tried. One interesting hypothesis towards explaining this observation is that to first approximation the data density for images is almost discrete, in the sense that what matters to get good images is whether you are pretty much on the data manifold (with a very small tolerance for noise) or off of it, and not so much the relative density on the manifold. This is simply a consequence of the manifold hypothesis stating that probability mass concentrates on the manifold.

---

> ### Author Response · Authors · 2018-11-23
> **Feedback by reviewer. Thanks for your time! :)**
>
> We would appreciate it if the reviewer could take another look at our changes, and let us know if the reviewer would like to request additional changes that would alleviate reviewers concerns. We hope that our updates  address the reviewer's concerns. We once again thank the reviewer for the feedback of our work.

---

### Official Review · AnonReviewer3 · 2018-11-05
**There are some unclear issues, regarding correctness and significance**

**Rating:** 5
**Confidence:** 5

**Review:**

It is well known that energy-based model training requires sampling from the current model.
This paper aims to develop an energy-based generative model with a generator that produces approximate samples.
For this purpose, this paper combines a number of existing techniques, including sampling in latent space, using a GAN-like technique to maximize the entropy of the generator distribution.
Evaluation experiments are conducted on toy 2D data, unsupervised anomaly detection, image generation.

The proposed method is interesting, but there are some unclear issues, which hurts the quality of this paper.

1. Correctness

The justification of adding a gradient norm regularizer in Eq. (3) for turning a GAN discriminator into an energy function is not clear.

Sampling in latent space and then converting to data space samples to approximate the sampling from p_theta is operationally possible. There are three distributions - the generator distribution p_G, the distribution p_comp implicitly defined by the latent-space energy obtained by composing the generator and the data-space energy, and the energy-based model p_E.
p_G is trained to approximate p_E, since we minimize KL(p_G||p_E). Does latent space sampling necessarily imply that p_comp leads to be closer to p_E ?

2. Significance

In my view, the paper is an extension of Kim&Bengio 2016.
Two extensions -  providing a new manner to calculate the entropy term, and using sampling in latent space. In this regard, Section 3 is unnecessarily obscure.

The results of image generation in Table 2 on CIFAR-10 are worse than WGAN-GP, which is now in fact only moderately performed GANs. In a concurrent ICLR submission - "Learning Neural Random Fields with Inclusive Auxiliary Generators", energy-based models trained with their method are shown to significantly outperform WGAN-GP.

---

> ### Author Response · Authors · 2018-11-14
> **Thanks for your feedback . Paper goes well beyond Kim & Bengio 2016**
>
> We thank the reviewer for their time and feedback. We  hope to address concerns the reviewer has here.
>
> * “The justification of adding a gradient norm regularizer in Eq. (3) for turning a GAN discriminator into an energy function is not clear.”
>
> Gradient norm regularizer in Eq. (3): the regularizer ||dEnergy(x)/dx||^2 is not just a smoothness regularizer but it also makes data points x energy minima (because ||dEnergy(x)/dx|| should be 0 at data points). This thus helps to learn a better energy function. Note that this is similar in spirit to score matching, which also carves the energy function so that it has local minima at the training points. The regularizer also stabilizes the temperature (scale) of the energy function, making training stable.
>
> * “In my view, the paper is an extension of Kim&Bengio 2016”
>
> We strongly believe this paper goes well beyond Kim & Bengio 2016. First, a major issue of Kim & Bengio 2016 is that it used covariance to maximize entropy. When we tried reproducing the results in that paper, even with the help of the authors, we could not get stable results. Entropy maximization using a mutual information estimator is much more robust compared to covariance maximization. But that alone was not enough and we got strong improvements by using the gradient norm regularizer (see (3) below) which helped stabilize the training as well. Finally, we show a successful form of MCMC exploiting the generator latent space composed with the energy function and we show new and successful empirical results on anomaly detection and sharp image generation, something which had not been done earlier for an energy-based model (and certainly not by Kim & Bengio). We also direct the reviewer towards our empirical results on discrete mode collapse where we show our model naturally covers all the modes in that data (in the expanded, 10^4 mode StackedMNIST dataset) and also better matches the mode count distribution as evidenced by the very low KL divergence scores.
>
> * “Sampling in latent space and then converting to data space samples to approximate the sampling from p_theta is operationally possible. There are three distributions - the generator distribution p_G, the distribution p_comp implicitly defined by the latent-space energy obtained by composing the generator and the data-space energy, and the energy-based model p_E., p_G is trained to approximate p_E, since we minimize KL(p_G||p_E). Does latent space sampling necessarily imply that p_comp leads to be closer to p_E ?”
>
> MCMC on the energy function p_E in data space did not give good results, while doing it in the latent space worked. We hypothesize that the reason for this is (a) walking on the data manifold is much easier in the latent space, as shown earlier by Bengio et al 2013 (because the data manifold has been somewhat flattened when represented in the latent space) and (b) composing the generator with the energy function gets rid of spurious modes of the energy which the generator cannot represent (if it did, then the negative samples from the generator would have made the energy function learn to get rid of these spurious modes, via the 2nd term of eqn 3 when training the energy function).
>
> * “The results of image generation in Table 2 on CIFAR-10 are worse than WGAN-GP, which is now in fact only moderately performed GANs. In a concurrent ICLR submission - "Learning Neural Random Fields with Inclusive Auxiliary Generators", energy-based models trained with their method are shown to significantly outperform WGAN-GP”
>
> The objective was not to beat the best GANs (which do not provide an energy function) but to show that it was possible to have both an energy function and good samples by appropriately fixing issues with the Kim & Bengio setup  (and we clearly did not know about the concurrent ICLR submissions on energy-based models).
>
>
> Please let us know if anything is unclear here or if there is any other comparison that would be helpful in clarifying things more.

---

> ### Author Response · Authors · 2018-11-23
> **Feedback by Reviewer. Thanks for your time! :)**
>
> We would appreciate it if the reviewer could take another look at our changes, and let us know if the reviewer would like to request additional changes that would alleviate reviewers concerns. We hope that our updates  address the reviewer's concerns about comparison to Kim and Bengio. We once again thank the reviewer for the feedback of our work.

---

> ### Comment · AnonReviewer3 · 2018-11-27
> **The method is problematic**
>
> The reviewer would like to thank the authors for their response, which clarifies some unclear issues. However, the response does not address my concern about the correctness of the proposed method.
>
> Particularly, p_EG(x) (the distribution implicitly defined by composing the generator and the data-space energy) != p_G(x).
> Note that the entropy term H(p_G) arises from using samples from p_G to approximate sampling from p_E in maximum likelihood training of p_E. If using samples from p_EG(x) for approximation, then the entropy term would be H(p_EG). Estimation of H(p_EG) is needed. The whole method seems to be problematic, from this perspective.
>
> Reviewer-2 shows the same concern.

---

> > ### Author Response · Authors · 2018-11-29
> > **Clarifications regarding MCMC**
> >
> > Yes, it is indeed correct that if we used samples from P_EG(x) to approximate samples from P_E, we need to be maximizing the entropy H(p_EG). In our particular case, we trained our models with n_mcmc = 0 (as mentioned in Section 7.2). Hence we used samples from P_G to approximate P_E (not P_EG).
> >
> > In short, we are NOT using MCMC in latent space (sampling from p_EG(x)) to produce the samples used to train the energy function E. We are using ordinary samples from p_G for this purpose, thus following equation 3 in the paper.  The latent-space MCMC in this version of our work is introduced as a sampling mechanism since MCMC in the visible space has issues with mixing and spurious modes. As mentioned in the rebuttals, the latent-space MCMC provides a way of performing MCMC sampling that hides the spurious modes of the energy function.

---

> > > ### Comment · AnonReviewer3 · 2018-11-29
> > > **This response make the paper more problematic.**
> > >
> > > > we trained our models with n_mcmc = 0 (as mentioned in Section 7.2)
> > >
> > > This further obscures the significance of section 4. And not good to state this in the appendix. Moreover, what is the evaluation result for the latent-space MCMC sampling?
> > >
> > > This response make the paper more problematic.
> > >
> > > The question in my first post regarding correctness is still not answered.

---

> > > > ### Author Response · Authors · 2018-11-30
> > > > **Evaluation results for MCMC sampling**
> > > >
> > > > > "This further obscures the significance of section 4. And not good to state this in the appendix. Moreover, what is the evaluation result for the latent-space MCMC sampling?"
> > > >
> > > > We have conducted additional experiments to address the concerns raised by the reviewer.Your feedback has already been very helpful in improving the paper. The reason we mentioned it in the appendix is that we presented our algorithm as a generic framework, and we cited what we did experimentally in this specific case in the appendix (experiments section).
> > > >
> > > > As per reviewer's request for evaluation results for MCMC sampling: we had originally only shown qualitative samples on CelebA. Upon your request, we decided to run quantitative numbers on CIFAR-10 and noticed that we get 6.50 (+- 0.6) Inception Score and 36.75 FID after applying multiple iterations of MCMC sampling.
> > > >
> > > > > "This response make the paper more problematic. The question in my first post regarding correctness is still not answered."
> > > >
> > > > If we understand right, the reviewer has raised 2 points regarding the correctness.
> > > > 1.) Justification for gradient norm regularizer.
> > > > 2.) Whether latent space sampling necessarily implies that P_EG leads to be closer to P_E
> > > >
> > > > 1.). Gradient norm regularizer in Eq. (3): the regularizer ||dEnergy(x)/dx||^2 is not just a smoothness regularizer but it also makes data points x energy minima (because ||dEnergy(x)/dx|| should be 0 at data points). This thus helps to learn a better energy function. Note that this is similar in spirit to score matching, which also carves the energy function so that it has local minima at the training points. The regularizer also stabilizes the temperature (scale) of the energy function, making training stable. Our diverse set of experiments on Anomaly Detection and Natural Image Generation corroborates this result.
> > > >
> > > >
> > > > 2.)  We admit that p_EG leads to be closer to P_E only if we minimize KL(P_EG || P_E), which requires us to use samples from P_EG to maximize entropy and train the Generator. This is a slight modification which should be made to the general framework (algorithm) presented in Section 4.
> > > >
> > > > In our specific paper, we mention that during training we minimize KL(P_G || P_E) [not KL(P_EG || P_E)]. As the reviewer points out, this means that latent space MCMC from P_EG does not mathematically imply that we sample from P_E. However, it is a beneficial / advantageous method of sampling from a region of P_E that hides the spurious modes of the energy function and provides better mixing.
> > > >
> > > > However, with the generic framework presented, it could well be made to sample from P_E, provided (like the reviewer mentioned), we minimize KL(P_EG || P_E).
> > > >
> > > > Thank you again for the thoughtful review. We would like to know if our response adequately addressed your concerns. Are there any other aspects of the paper that you think could be improved? Are there particular additional methods that the reviewer would prefer a comparison to? Or anything else we can do to address the concern about the motivation? We think these additional analyses answers reviewer's questions about the novelty and the application of gradient norm regularizer. Does reviewer still think, that it's problematic ? If yes, then can reviewer be precise what the reviewer has in mind ?
> > > >
> > > > Our work proposed a way to begin addressing these challenges, and compares extensively to prior papers and several ablations. Specifically, we tackle an important problem of  training energy-based models with maximum likelihood, by providing a way to approximate the negative phase using a generator, which is trained using neural estimators of mutual information to approximate the entropy and a score norm penalty that is very crucial for training stability. We also provide a general, flexible framework where MCMC in latent-space could be used to further better approximate the negative phase and produce better samples which mix substantially better than visible space MCMC. We corroborate our claims through a diverse set of experiments on toy data, discrete mode collapse, natural image generation and anomaly detection and show that our results are very encouraging since our model does not exhibit mode dropping, produces sharp images and produces state of the art results on anomaly detection among energy-based models.

---

### Official Review · AnonReviewer1 · 2018-11-06
**An interesting combination of the recently developed techniques for a better algorithm**

**Rating:** 6
**Confidence:** 4

**Review:**


In this paper, the authors extend the framework proposed by Kim&Bengio 2016 and Dai et.al. 2017, which introduce an extra step to fit a generator to approximate the current model for estimating the deep energy model. Specifically, the generator is fitted by reverse KL divergence. To bypass the difficulty in handling the entropy term, the authors exploit the Deep INFOMAX formulation, which introduces one more discriminator. Finally, to obtain better samples, the authors inject the Metropolis-adjusted Langevin algorithm within the learned generator to generate samples in latent space. They demonstrate the better performances of the proposed algorithms in both synthetic and real-world datasets, and apply the learned model for anomaly detection task.

The paper is well-written and does a quite good job in combining several existing algorithms to obtain the ultimate algorithm. The algorithm achieves quite good empirical performances. However, the major problem of this paper is the novelty. The algorithm is basically an extension of the Kim&Bengio 2016 and Dai et.al. 2017, with other existing learning technique. Maybe the only novel part is combining the MCMC with the learned generator for generating samples. However, the benefits of such combination is not well justified empirically. Based the figure 4, it seems the MCMC does not provide better samples, comparing to directly generate samples from G_z. It will be better if the authors can justify the motivation of using MCMC step.

Secondly, it is reasonable that the authors introduce the gradient norm as the regularization to the objective for training stability. However, it will be better if the effect of the regularization for the energy model estimation can be discussed.

Minor:
The loss function for potential in Eq(3) is incorrect and inconsistent with the Algorithm 1. I think the formulation in the Algorithm box is correct.

In sum, I personally like the paper as a nice combination of recently developed techniques to improve the algorithm for solving the remaining problem in statistics. The paper can be better if the above mentioned issues can be addressed

---

> ### Author Response · Authors · 2018-11-14
> **Clarifying technical questions and providing justification**
>
> We thank the reviewer for their time and feedback. We hope to address concerns the reviewer has here.
>
> * “However, the major problem of this paper is the novelty. The algorithm is basically an extension of the Kim&Bengio 2016 and Dai et.al. 2017, with other existing learning technique”
>
> We strongly believe this paper goes well beyond Kim & Bengio 2016. First, a major issue of Kim & Bengio 2016 is that it used covariance to maximize entropy. When we tried reproducing the results in that paper, even with the help of the authors, we could not get stable results. Entropy maximization using a mutual information estimator is much more robust compared to covariance maximization. But that alone was not enough and we got strong improvements by using the gradient norm regularizer (see (3) below) which helped stabilize the training as well. Finally, we show a successful form of MCMC exploiting the generator latent space composed with the energy function and we show new and successful empirical results on anomaly detection and sharp image generation, something which had not been done earlier for an energy-based model (and certainly not by Kim & Bengio). We also direct the reviewer towards our strong empirical results on discrete mode collapse where we show our model naturally covers all the modes in that data (in the expanded, 10^4 mode StackedMNIST dataset) and also better matches the mode count distribution as evidenced by the very low KL divergence scores.
>
>
> * “Maybe the only novel part is combining the MCMC with the learned generator for generating samples. However, the benefits of such combination is not well justified empirically. Based the figure 4, it seems the MCMC does not provide better samples, comparing to directly generate samples from G_z. It will be better if the authors can justify the motivation of using MCMC step.”
>
> Our contribution is also to enforce entropy maximization on the discriminator and using a regularizer on the energy-function to stabilize training. This specific combination was instrumental in obtaining our empirical result: (1) Covering all modes in our discrete mode collapse experiment where our model matches the mode count distribution of the data significantly better than WGAN-GP as pointed out in Section 5.2 and Table 1. (2) Using the learned energy function to perform anomaly detection, beating the previous SOTA energy-based model (DSEBM) by a large margin (as mentioned in Section 5.4 Table 3) and comparable to the SOTA anomaly detection method (DAGMM) which is purely designed for anomaly detection and not generative modeling (3) Natural image generation, where our energy-based method performs comparable to a strong WGAN-GP baseline in perceptual quality and doesn’t exhibit the common blurriness issue in standard maximum-likelihood trained EBMs (Section 5.3 Table 2).
>
> Regarding the justification of latent space MCMC: Note that the MCMC on the energy function in data space did not give good results, while doing it in the latent space worked. (Refer Figure 5 for data-space MCMC samples). We hypothesize that the reason for this is (a) walking on the data manifold is much easier in the latent space, as shown earlier by Bengio et al 2013 and (b) composing the generator with the energy function gets rid of spurious modes of the energy which the generator cannot represent (if it did, then the negative samples from the generator would have made the energy function learn to get rid of these spurious modes).
>
> * “Secondly, it is reasonable that the authors introduce the gradient norm as the regularization to the objective for training stability. However, it will be better if the effect of the regularization for the energy model estimation can be discussed.”
>
> Effect of the regularization of the energy function: the regularizer ||dEnergy(x)/dx||^2 is not just a smoothness regularizer but it also makes data points x energy minima (because ||dEnergy(x)/dx|| should be 0 at data points). This thus helps to learn a better energy function. Note that this is similar in spirit to score matching, which also carves the energy function so that it has local minima at the training points (i.e it is helping to make data points as an energy minima). The regularizer also stabilizes the temperature (scale) of the energy function, making training stable.
>
> * “The loss function for potential in Eq(3) is incorrect and inconsistent with the Algorithm 1. I think the formulation in the Algorithm box is correct.”
>
> Indeed there was a typo in eqn 3. The LHS should have been the gradient of L_E wrt theta, and Omega on the RHS should have been dOmega/dtheta.

---

> ### Author Response · Authors · 2018-11-23
> **Feedback by Reviewer. Thanks for your time! :)**
>
> We would appreciate it if the reviewer could take another look at our changes, and let us know if the reviewer would like to request additional changes that would alleviate reviewers concerns. We hope that our updates  address the reviewer's concerns. We once again thank the reviewer for the feedback of our work.

---

### Public Comment · ~Xiaojian_Ma1 · 2018-09-28
**Good paper with insightful ideas**

Previously I've read a NIPS workshop paper[Finn et.al., 2016] that try to reveal the inherent connection between training an energy-based model and a generative adversarial net. The main contribution of that paper is providing a full derivation of the equivalence of $L_G \leftrightarrow KL(p_G || p_\theta)$ and $L_D \leftrightarrow \mathbb{E}_{x \sim p_{data}}\left[-\log{p_\theta(x)}\right]$. However, in that paper, this equivalence only holds when D takes the form of $\frac{p_\theta(x)}{p_\theta(x) + p_G(x)}$, and this is so-called the "optimal discriminator" mentioned in [Goodfellow et.al., 2014]. But the problem is the discriminator actually cannot always hold such form during gradient descent, which implies that it's basically not appropriate to directly cast the training of original GAN as training an EBM as what [Finn et.al.,2016] claimed.

In this paper, the authors alternatively choose to optimize $KL(p_G || p_\theta)$ in Eq.4 by explicitly maximizing the entropy of G with DIM estimator, such techniques eliminate the dependency of the optimal discriminator in [Finn et.al.,2016] while the equivalence to the EBM objective could still be retained. On the other hand, although the proposed method still relies on MCMC, sampling with learned energy could be an essence to optimizing strictly with the EBM objective in this generator-discriminator architecture, and the authors do report promising results compared with original GAN and WGAN-GP.
([Finn et.al., 2016] tries to prove that the original GAN training procedure implicitly contains the MCMC step for estimating the partition function, but such conclusions depend on the optimal discriminator form).

One minor suggestion, Is there a typo in Sec.3? I think KL(p_||p_\theta) = H[p_G] - E_{p_G}[log pθ(x)] should be KL(p_||p_\theta) = -H[p_G] - E_{p_G}[log pθ(x)], a minus is missing.

[Finn et.al., 2016] A Connection between Generative Adversarial Networks, Inverse Reinforcement Learning, and Energy-Based Models, in NIPS Workshop, 2016, https://arxiv.org/abs/1611.03852
[Goodfellow et.al., 2014] Generative Adversarial Nets, in NIPS, 2014

---

> ### Author Response · Authors · 2018-09-30
> **Thanks for the feedback and spotting the typo**
>
> Jeasine, thanks for the feedback and spotting the sign typo! We will add the very relevant Finn et al 2016 reference.

---

### Public Comment · (anonymous) · 2018-10-02
**Some questions on the details**

This paper proposed an interesting idea on combining the GANs and energy based models.
I have following questions on the details of this paper:
1) In Equation (3), there is a regularization item added to avoid the "numerical problems". It looks very similar to the Gradient Penalty item  in WGAN-GP[Gulrajani et al., 2017]. In WGAN-GP, the gradient norm regularizer is utilized on the linear interpolation space of P_D and P_G. While in algorithm 1 of this paper, I find the regularizer only penalized the point on real data distribution, and second term in Equation(3) can be infinity as  there is no constraint on P_G.  I think that this will still cause unstable training and numerical problems.

2) To make P_\theta and P_G match, the authors selected to minimize a KL divergence. As illustrated in [Arjovsky et al., 2017], KL divergence will behave poorly when the two distribution is without overlapping. I am curious about whether the KL divergence is suitable to estimate the difference on P_\theta and P_G,  especially when the distribution is high dimensional.

---

> ### Author Response · Authors · 2018-10-02
> **Clarifications on the details**
>
> Thanks for your questions.
> 1.) Yes, as you point out, we noticed as well that our score norm penalty is similar to the WGAN-GP regularizer. Our intuition behind using the score penalty regularizer in our case is to have 0 score norm for training points because that corresponds to a minimum of the energy (and we want to carve the energy function to have minima at data points). That is, there shouldn't be any gradient with respect to the input for true data (or equivalently, 0 reconstruction error). This, in practice was sufficient to fix the temperature explosion issue as mentioned in Section 3. We also noticed that in practice, the absence of constraint on P_G (fake samples) does not cause unstable training or any numerical problems.
>
> 2.) Yes, it is right to point out that if the distributions of P_\theta and P_G do not have overlapping support, the KL divergence will be infinite and there will be no gradient signal to align the 2 distributions. However, our intuition is that this is less likely to happen since the entropy maximization term arising from the KL divergence will help align the supports of the 2 distributions and hence provide gradient signal to match the 2 distributions. Also, the KL gradient on P_G is basically doing two reasonable things: putting more probability mass where the energy is low and increasing entropy (otherwise all the mass could be concentrated in one point). Note how the latter term will prevent p_G to be too concentrated.

---

### Meta-Review · Area_Chair1 · 2018-12-10
**incremental extension of past work; benefits unclear**

**Confidence:** 5
**Recommendation:** Reject

**Metareview:**

The proposed method is an extension of Kim & Bengio (2016)'s energy-based GAN. The novel contributions are to approximate the entropy regularizer using a mutual information estimator, and to try to clean up the model samples using some Langevin steps. Experiments include mode dropping experiments on toy data, samples from the model on CelebA, and measures of inception score and FID.

The paper is well-written, and the proposal seems sensible. But as various reviewers point out, the work is a fairly incremental extension of Kim and Bengio (2016). Most of the new elements, such as Langevin sampling and the gradient penalty, have also been well-explored in the deep generative modeling literature. It's not clear there is a particular contribution here that really stands out.

The experimental evidence for improvement is also fairly limited. Generated samples, inception scores, and FID are pretty weak measures for generative models, though I'm willing to go with them since they seem to be standard in the field. But even by these measures, there doesn't seem to be much improvement. I wouldn't expect SOTA results because of computational limitations, but the generated samples and quantitative evaluations seem worse than the WGAN-GP, even though the proposed method includes the gradient penalty and hence should be able to at least match WGAN-GP. The MCMC sampling doesn't appear to have helped, as far as I can tell.

Overall, the proposal seems promising, but I don't think this paper is ready for publication at ICLR.